# Ibogaine Induces Cardiotoxic Necrosis in Rats—The Role of Redox Processes

**DOI:** 10.3390/ijms25126527

**Published:** 2024-06-13

**Authors:** Teodora Vidonja Uzelac, Nikola Tatalović, Milica Mijović, Marko Miler, Tanja Grahovac, Zorana Oreščanin Dušić, Aleksandra Nikolić-Kokić, Duško Blagojević

**Affiliations:** 1Department of Physiology, Institute for Biological Research “Siniša Stanković”—National Institute of the Republic of Serbia, University of Belgrade, Bulevar Despota Stefana 142, 11060 Belgrade, Serbia; teodora.vidonja@ibiss.bg.ac.rs (T.V.U.); tanja.grahovac@ibiss.bg.ac.rs (T.G.); zoranaor@ibiss.bg.ac.rs (Z.O.D.); san@ibiss.bg.ac.rs (A.N.-K.); 2Institute of Pathology, Faculty of Medicine, University of Priština, Anri Dinana bb, 38220 Kosovska Mitrovica, Serbia; milica.mijovic@med.pr.ac.rs; 3Department of Cytology, Institute for Biological Research “Siniša Stanković”—National Institute of the Republic of Serbia, University of Belgrade, Bulevar Despota Stefana 142, 11060 Belgrade, Serbia; marko.miler@ibiss.bg.ac.rs

**Keywords:** ibogaine, noribogaine, heart, cardiotoxicity, myocardial necrosis, vasculitis, pericarditis, antioxidant defense, sex-associated differences

## Abstract

Ibogaine is an organic indole alkaloid that is used in alternative medicine to combat addiction. Numerous cases of life-threatening complications and sudden deaths associated with ibogaine use have been reported, and it has been hypothesized that the adverse effects are related to ibogaine’s tendency to induce cardiac arrhythmias. Considering that the bioavailability of ibogaine and its primary metabolite noribogaine is two to three times higher in female rats than in male rats, we here investigated the effect of a single oral dose (1 or 20 mg/kg) of ibogaine on cardiac histopathology and oxidative/antioxidant balance. Our results show that ibogaine induced dose-dependent cardiotoxic necrosis 6 and 24 h after treatment and that this necrosis was not a consequence of inflammation. In addition, no consistent dose- and time-dependent changes in antioxidant defense or indicators of oxidative damage were observed. The results of this study may contribute to a better understanding of ibogaine-induced cardiotoxicity, which is one of the main side effects of ibogaine use in humans and is often fatal. Nevertheless, based on this experiment, it is not possible to draw a definitive conclusion regarding the role of redox processes or oxidative stress in the occurrence of cardiotoxic necrosis after ibogaine administration.

## 1. Introduction

Ibogaine (PubChem CID: 197060) is an organic heteropentacyclic compound, the most abundant alkaloid in the root bark of the rainforest shrub iboga (*Tabernanthe iboga* Baill.), which grows in West Africa. It is traditionally used by local communities to overcome fatigue, hunger, and thirst, and it is used in higher doses to induce hallucinations during spiritual rituals [1]. Reports on the effects of ibogaine on human health are inconclusive [2,3], but it is used in alternative medicine, which includes a so-called “ibogaine medical subculture”, as a means to combat addiction [4,5]. It is widely touted as an effective treatment against addiction to a range of substances, including opioids, cocaine, amphetamines, alcohol, marijuana, and nicotine; as a treatment for many mental and emotional disorders; and as a “powerful tool for self-discovery and personal growth”. However, numerous cases of life-threatening complications and sudden deaths associated with ibogaine use have been reported [2,6,7]. Some of the deaths occurred several days after ingestion or after ingestion of very small doses [8]. It has been hypothesized that the adverse effects are related to the propensity of ibogaine to induce cardiac arrhythmias [6]. Numerous cases of ibogaine-induced QT interval prolongation and associated life-threatening torsade de pointes (TdP) arrhythmias have also been reported in individuals without preexisting cardiovascular disease or family history of cardiovascular disease [3]. Furthermore, female sex is among the additional risk factors for drug-induced TdP arrhythmias that should be considered before the use of ibogaine [6]. Previous results have shown that the bioavailability of ibogaine and its principal in vivo metabolite, noribogaine, is two to three times higher in female rats than in male rats [9,10] and that ibogaine has sex-specific effects on the central nervous system [11,12,13], liver, and kidneys, with ibogaine-induced pathological changes being more pronounced in female rats [10,14,15].

It has been shown in several in vitro, ex vivo and in vivo experimental models that the biological activity of ibogaine, in addition to involving interaction with various types of receptors [2,16,17], is caused by rapid depletion of ATP, which is followed by an induction of enzymes related to energy metabolism, as well as increases in cellular reactive oxygen species (ROS) and the activity of antioxidant enzymes [18,19,20,21,22,23,24]. In our previous experiments in rats treated per os it was shown that ibogaine affected energy metabolism, antioxidant defense, and redox balance without life-threatening pathological effects on the liver and kidneys [10,14,15].

Considering the pharmacokinetics of ibogaine, adverse cardiac effects and sex-based differences, as well as its influence on energy metabolism, ROS and antioxidant enzyme activity, as studied in various experimental models in vitro [21], ex vivo [19,20,21,22,23], and in vivo [10,14,15], we here present the effects of ibogaine on the hearts of male and female Wistar rats in vivo 6 and 24 h after treatment with a single oral dose of 1 or 20 mg/kg body weight (b.w.) (human-equivalent doses based on body surface area of 0.16 or 3.2 mg/kg b.w., respectively). Ibogaine doses were selected according to the most common doses used to facilitate abstinence from a variety of addictive drugs, suggestions from the Global Ibogaine Therapy Alliance (GITA), and doses in clinical trials approved by the U.S. Food and Drug Administration (FDA) Advisory Panel, as well as previous results from laboratory-animal studies [10,14,15]. Because of the cardiotoxicity of ibogaine, histologic analyzes were performed to investigate possible pathologic changes in cardiac tissue morphology. Since irregularities in cardiac glycogen metabolism are common manifestations of various cardiopathologies [25] and considering the effects of ibogaine on energy metabolism, in this experiment, we examined the amount of glycogen in cardiomyocytes as a potential indicator of cardiac metabolic stress and glucose metabolism. Bearing in mind that ibogaine’s mechanisms of action include ROS production and considering the role of oxidative stress in heart disease and the influence of redox imbalance on cardiac pathology [26,27], we also examined the effects of ibogaine on oxidative/antioxidative balance in rat hearts by measuring the activity of the following antioxidant enzymes: cytosolic CuZn superoxide dismutase (SOD1), mitochondrial Mn superoxide dismutase (SOD2), catalase (CAT), glutathione peroxidase (GSH-Px), and glutathione reductase (GR). Glutathione S-transferases (GST) and xanthine oxidase (XOD) activities were also measured as indicators of CYP-related metabolism and purine metabolism. The intensity of possible ROS-mediated lipid peroxidation was estimated by measuring TBARS concentration. The concentration of free protein and nonprotein sulfhydryl groups (–SH), i.e., thiols, was measured as an indicator of a thiol-based redox state.

## 2. Results

### 2.1. Histopathological Analysis

Treatment with ibogaine induced histopathological changes in cardiac tissue in both male and female rats (Figure 1a, Table 1a), with myocardial necrosis being the most prominent change. Myocardial necrosis was present in the cardiac-tissue samples from at least some, if not all, animals in each treated group. In both male and female rats, a lower dose resulted in focal necrosis, whereas a higher dose resulted in diffuse necrosis. Myocardial mononuclear cell infiltrate was absent from all samples, suggesting that necrosis of cardiomyocytes was not a consequence of inflammation, i.e., myocarditis. On the other hand, a perivascular inflammatory infiltrate, an indicator of vasculitis, was observed almost exclusively in male rats treated with a lower dose. By contrast, it was predominant in female rats treated with a higher dose. Pericarditis occurred only in female rats 24 h after treatment with a lower dose of ibogaine. The amount of adipose tissue remained unchanged in all treated groups.

### 2.2. Presence of Glycogen and Color Intensity of Periodic Acid-Schiff (PAS) Staining

Our results showed that the intensity of PAS staining was low in all animals except one untreated female rat. The prevalence of glycogen-positive cells in heart-tissue samples from untreated male rats was less than 33.3% (Figure 1b, Table 1b). Treatment with a lower dose of ibogaine increased this percentage to more than 33.3%, but the phenomenon was recorded after 6 h in all rats and after 24 h only in two of four animals. After treatment with a higher dose, the percentage of glycogen-positive cells was the same as in the control group. In untreated female rats, the prevalence of glycogen-positive cells was higher than in males, i.e., more than 33.3% or even more than 66.6%. Twenty-four hours after treatment with the lower dose and both 6 and 24 h after treatment with the higher dose, the percentage of glycogen-positive cells in the heart-tissue samples of the female rats had decreased compared to the percentage in the controls.

### 2.3. Enzyme Activities and Concentration of TBARS and –SH Groups

In male rats (Table 2, Table 3a), 6 h after treatment with a lower dose of ibogaine, TBARS concentration had significantly increased (Tuckey’s HSD test, *p* < 0.05), while at the same time, the concentration of free protein –SH groups had decreased (*p* < 0.05). Six hours after treatment with a higher dose, the activities of SOD2 and GST had decreased (*p* < 0.001 and *p* < 0.05, respectively). In female rats treated with a higher dose of ibogaine (Table 2, Table 3b), the activities of SOD1 and GR in the cardiac tissue had significantly decreased after 24 h (*p* < 0.01 and *p* < 0.001, respectively).

To test for possible sex-associated differences and evaluate the general trend in changes induced by ibogaine treatment, we performed three-way ANOVA using sex, dose, and time after treatment as factors. In general, our analysis showed (Figure 2, Table 4) that the degree of changes in the activities of SOD1, GSH-Px, GR, and XOD, as well as in the levels of TBARS and the nonprotein –SH groups, were strongly sex-dependent: females responded to ibogaine with decreases in SOD1, GR, and XOD activities, as well as with decreases in the concentrations of non-protein –SH groups, while males showed elevated GSH-Px activity and TBARS concentrations. Dosage was also shown to be a significant factor in the changes in the activities of the enzymes SOD1, CAT, GR, and GST: a higher dose of ibogaine led to lower activities than did a lower dose. Moreover, time after ibogaine application was also a significant factor leading to increased SOD1, GSH-Px, and GR activity; values were higher at 24 h after treatment than at 6 h. In contrast, the concentration of the nonprotein –SH groups was higher at 24 h than at 6 h.

## 3. Discussion

The most important finding of our research is the presence of myocardial necrosis after ibogaine application. The dose-dependence of the ibogaine effect was evident, as the higher dose resulted in more extensive, i.e., more diffuse, necrosis. Lower doses of ibogaine resulted in focal necrosis or lack of necrosis in some animals, which were examined histologically on the lower left ventricular wall. The most common sites for focal lesions are the left ventricular papillary muscles and the subendocardial myocardium [28], so based on the results of our experiment, we cannot exclude the possibility that at least focal necrosis was present in the myocardium of all treated animals. Necrosis is associated with ATP depletion, as well as with extensive ROS production [29]; thus, the ibogaine dose–response in terms of the presence and extent of necrosis could be the consequence of the extent of both ATP depletion and the generation of ROS, particularly hydrogen peroxide, and vice versa. It has been shown that both low and high concentrations of H_2_O_2_ can trigger cell death through mechanisms such as ATP depletion or a decrease in cellular GSH [30]. However, the necrosis levels observed in our work were not directly indicative of a dose-response relationship, which would be associated with a large reduction in antioxidant activity or GSH, and the antioxidant defense response observed in the hearts did not follow the ibogaine dose-response pattern. Furthermore, the antioxidant response is highly sex-specific and the only common change was a decrease in GR activity, indicating impaired reduction of cellular glutathione and its impaired turnover as a consequence; this change could lead to a lower amount of reduced glutathione being available to prevent necrosis. On the other hand, necrosis may be prevented by a strong and rapid increase in antioxidant enzymes at the onset of the ROS burst, which sets the stage for apoptosis [31]. In our experiment, the histologic examination was performed on a small part of the heart, namely the caudolateral part (lower left part) of the left ventricular wall, and histochemical staining with hematoxylin and eosin was performed to detect histopathologic changes. Using this method, we could not detect any signs of massive apoptosis, e.g., eosinophilic aggregates, which would indicate the massive presence of apoptotic corpuscles. The presence of necrosis in the hearts of ibogaine-treated rats demonstrates that the ibogaine-induced ROS burst may be an early event and that the changes in antioxidant enzyme activity detected at both 6 and 24 h may represent the late phase, i.e., the consequences of the immediate effects of ibogaine. However, ROS concentrations were not measured in our experiment. Our previous results have shown that the increase in antioxidant enzyme activity indeed occurs only a few hours after ibogaine treatment and can be prevented by inhibition of cellular signaling pathways involving ROS, namely mitochondrial K_ATP_ channels and β1-adrenergic receptors [22,23]. This also suggests involvement in ROS-mediated cellular transduction and mediation of ibogaine effects, albeit as early effects. There are data suggesting that activation of mitochondrial K_ATP_ channels protects against necrosis and apoptosis in cardiac myocytes [32] and that overexpression of cyclic GMP-dependent protein kinases attenuates necrosis and apoptosis after ischemia/reoxygenation in adult rat cardiomyocytes via ROS generation and mitochondrial K_ATP_ channel opening [33]. Mitochondrial K_ATP_ channel opening occurs when ATP is not present to block it and occurs only in circumstances of ATP depletion (and/or ADP accumulation) [34]. Furthermore, the interplay between mitochondrial K_ATP_ channels and β-adrenergic receptors represents an important signaling pathway that directs both cellular and overall cardiac fate [35].

Despite the myocardial necrosis that occurred in most of the treated rats, no inflammatory processes in the myocardial tissue (no infiltrate of inflammatory cells between the cardiomyocytes) was observed using H&E staining, suggesting that administration of ibogaine caused acute myocardial cell necrosis without associated inflammation. However, changes in immune-cell infiltration cannot be excluded because no flow cytometric analysis for immune cells was performed. Typically, acute cardiac lesions are characterized by a scattered pattern of degeneration of small groups of cells rather than by the coagulation necrosis of a region of the myocardium, as seen in ischemic injury. These lesions are referred to as “myocytolysis”. The cellular damage in this case is not due to reduced perfusion, but rather to a direct biochemical or metabolic mechanism of cardiomyocyte injury [36,37]. Cardiotoxic necrosis is an irreversible process triggered by a severe disruption of cardiomyocyte homeostasis through any of several mechanisms, including high levels of ROS, defective Ca^2+^ signal transduction or inhibition of protein synthesis. The agents that most commonly cause acute cardiomyocyte cell death belong to the category of sympathomimetics, e.g., cocaine, but many other agents such as catecholamines, interleukin-2, cyclophosphamide, emetine, lithium, phenothiazines, antimony, and arsenicals have also been associated with acute myocyte death [36]. In addition to myocardial necrosis, administration of ibogaine led to the development of vasculitis in both sexes, with the difference that it occurred mainly after administration of a lower dose in males and after administration of a higher dose in females, suggesting a sex-based difference. In the females, three out of four animals showed signs of pericarditis 24 h after administration of the lower dose. It appears that the lower dose induced some inflammatory response, but not at the level of the myocardial tissue. This result also demonstrates that the effects of ibogaine differ not only in their intensity but also in their impacts on different tissues.

Our previous experiments also showed sex-specific histopathological changes in the livers and kidneys of rats. In female rats, ibogaine caused dilatation of the central vein and a small branch of the portal vein, whereas no histopathologic changes were observed in the livers of male rats. In the kidneys, pathological changes at the level of proximal tubules and tubular epithelial cells also appeared to be slightly more pronounced in female rats. In conjunction with the histopathologic observations in rat hearts, these differences might be additionally attributed to differences in the bioavailability of ibogaine and noribogaine between the sexes. The concentrations of ibogaine and noribogaine in the blood plasma were higher in female rats than in male rats [10,14], (Appendix A Table A1), as was the bioavailability [9]. Due to its lipophilic nature, ibogaine accumulates in adipose tissue and the brain and is slowly released into the blood. In the blood, ibogaine is metabolized to noribogaine and rapidly cleared [38], but elimination of noribogaine is slower [2]. Twenty-four hours after administration per os, 65% of ibogaine is excreted in the urine and feces in rats [9], and in humans, this figure is as high as 90% [39]. Our results in hearts showed no sex difference in the propensity of ibogaine to induce myocardial necrosis. It appears that under our experimental conditions, a two- or threefold difference in bioavailability was not sufficient to produce observable differences in the induction of necrosis, while on the other hand, a twentyfold difference in the administered dose produced obvious differences in the intensity of necrosis.

Regarding the effects of ibogaine on metabolism at the cardiac level, our results showed that the distribution of glycogen granules differed according to sex, i.e., males had a lower percentage of glycogen-positive cells than did females, but the total amount of glycogen was estimated to be low in both sexes, as indicated by the intensity of PAS staining. In males, a low dose of ibogaine increased the percentage of glycogen-positive cells. In females, the shift was reversed, with the low dose associated with fewer glycogen-positive cells. Glycogen takes up about 2% of the cell volume in an adult cardiomyocyte [40]. In contrast to liver and skeletal muscle, glycogen levels in cardiac muscle are elevated during fasting. This observation is consistent with the general principle that fatty acids, the predominant fuel for the heart during fasting, inhibit glycolysis more than they inhibit glucose uptake, diverting glucose to glycogen synthesis [41,42,43,44]. Excessive glycogen accumulation in cardiomyocytes may occur due to some chronic diseases or to prolonged treatment with drugs such as dexamethasone or toxins, leading to deterioration of cardiac function with manifestations such as arrhythmias, ventricular hypertrophy, and subsequent heart failure [44,45,46,47]. In view of this, it appears that males, unlike females, respond to some extent to cellular cardiac fasting. On the other hand, glycogen is rapidly depleted during adrenergic stimulation or a sudden increase in cardiac work [43], which may be the case in females. However, following ibogaine exposure, there were no changes in total glycogen levels; instead, the redistribution of glycogen prevented disruption of cardiac energetic homeostasis. Considering our previous results [10,14], which showed a decrease in glycogen content in the livers of rats treated with ibogaine at the same dose and for the same period of time as in the present work, we can argue that the systemic glycogen deprivation induced by ibogaine was not intense enough to induce an increase in total glycogen content in the myocardium. Furthermore, the changes in the number of glycogen-positive cells in our experiment could not be correlated with the amount of food ingested or the glucose concentration in the blood (see Table A2 in the Appendix A). The differences between male and female rats could also be due to the fact that estrogen acts on important glycogen-regulating proteins [25].

In contrast to the induction of myocardial necrosis, which was the same in both sexes, ibogaine treatment appears to have led to subtle differences in the degree of adaptation of antioxidant enzyme activity to ROS perturbations and redox changes. In our experiment, oral treatment with a single dose of ibogaine resulted in significant changes in the activity of some antioxidant enzymes and in the concentrations of TBARS and free protein –SH groups in the hearts of male and female rats, suggesting an imbalance in redox homeostasis. In our previous experiments, the effects of ibogaine on antioxidant enzymes were found to be tissue-specific, sex-specific, and dose- and time-dependent [10,14,15]. The changes observed in the heart are also characterized by some temporal dynamics and by both dose- and sex-associated differences, but the effects of ibogaine do not uniformly follow the time and dose patterns. Higher dose and longer exposure time did not lead to clear indicators of higher oxidative stress, but rather to subtle changes in ROS and redox homeostasis and a subsequent response in antioxidant enzyme activity. Only a lower dose of ibogaine induced a higher TBARS concentration and a lower concentration of free protein –SH groups in male rats after 6 h, indicating a state of oxidative stress, i.e., increased lipid peroxidation and oxidation of protein-bound –SH groups. Lipid peroxidation may impair cardiac signal transduction and affect cardiac contractility, while increased oxidation of protein-bound cysteine residues indicates a possible non-homeostatic redox regulatory state. After 24 h, overall ROS and redox homeostasis appear to be established in males treated with 1 mg/kg b.w. ibogaine, as no changes in the level of antioxidant enzyme activity were observed. However, a higher dose resulted in a significant decrease in SOD2 and GST activity in males after 6 h, suggesting that mitochondrial and cytoplasmic metabolic-detoxification processes were underway. Hydrogen peroxide can inhibit SOD2 [48,49], which could be interpreted as a potential increase in ROS, but without evidence of lipid peroxidation or a decrease in –SH groups. Taken together, these results suggest that the multifaceted nature and pharmacological properties of ibogaine operate at different cellular levels that may fall under the umbrella of a general mechanism but are specific to each dose and time point studied. In females, despite higher concentrations of ibogaine and noribogaine present in the circulation, no changes in the activity of antioxidant enzymes were observed after application of a lower dose. Only a higher dose resulted in a decrease in SOD1 and GR activity in the hearts of females after 24 h, suggesting a state of oxidative stress and impaired redox homeostasis in the cytoplasm due to superoxide accumulation and lower glutathione turnover. In both female and male animals, a higher dose of ibogaine led to necrosis in all treated animals, which may also be a consequence of low antioxidant activity.

Mechanisms of cardiotoxicity generally include (1) exaggerated pharmacologic effects of drugs that act on cardiovascular tissues, (2) exposure to substances that impair myocardial function, (3) direct damage to cardiomyocytes by chemicals, or (4) hypersensitivity reactions. [28]. Since the cardiotoxicity of ibogaine is well established in the literature, it has been recommended that ibogaine should only be allowed to be used by trained medical personnel under strict medical observation and continuous electrocardiographic monitoring over an extended period of time, with careful consideration of the longevity of noribogaine in human plasma. Furthermore, additional risk factors for drug-induced TdP arrhythmias must be clarified prior to the use of ibogaine [3,6]. The focus of research in this area has been the potassium channels of the ether-a-go-go-related gene (hERG) in the heart. These channels play an important role in the repolarization phase of cardiac action potentials, and blockade by ibogaine delays this repolarization, leading to prolongation of the QT interval and subsequently to arrhythmias and sudden cardiac arrest [3]. It has also been hypothesized that sudden cardiac death after ibogaine ingestion may be related to dysregulation of the autonomic nervous system. Namely, low doses of ibogaine stimulate the sympathetic nervous system, whose overactivity can lead to fatal cardiac arrhythmias. In contrast, high doses lead to a dominance of the parasympathetic nervous system, which could have a protective effect on the cardiovascular system. This hypothesis could explain the adverse effects of ibogaine administered at low doses and the adverse effects that occur long after the maximum plasma concentrations of the high doses have been reached [8].

The results of our experiment suggest that myocardial necrosis represents a new potential mechanism of ibogaine cardiotoxicity, namely the induction of cardiac arrhythmias and sudden cardiac death. The consequences of myocardial necrosis vary depending on the extent of the damage and its location. Sudden death due to acute heart failure may occur if the damage to the myocardium is too extensive. Necrosis-related cardiac arrhythmias may occur and even lead to sudden death if the conduction of the heart is disturbed by the necrosis. The response to conduction-system injury is poorly documented because histopathologic examination of the conduction system is labor-intensive and rarely performed. In the rare cases in which histopathologic and electrocardiographic studies have been available, the atrial and/or ventricular origin of the arrhythmia has been associated with inflammation, degeneration, and fibrosis along the cardiac conduction system [28]. Hypothetically, even focal necrosis caused by a lower dose of ibogaine could lead to a fatal outcome, depending on its location. This experiment used young and healthy rats that presumably had no cardiovascular disease and were therefore not particularly susceptible to adverse cardiovascular effects of ibogaine. Despite the development of myocardial necrosis, all rats in this experiment survived treatment for 6 or 24 h without any observed sign of health deterioration. However, the question remains as to how the health of their cardiovascular system would have developed over a longer period of time. There is also the question of how ibogaine affects rats that already have certain pathological conditions of the cardiovascular system or rats that have been exposed to opiate use. It should be kept in mind that this study was conducted as a single experiment, but diffuse myocardial necrosis occurred in all groups treated with the higher dose, regardless of sex or duration of treatment. On the other hand, we focused only on the differences in the observed changes in antioxidant enzyme activities with high statistical significance. Because the cardiac effects of ibogaine in humans are poorly understood, animal models are one of the best surrogates for human studies, with the caveat that the pathologic and biochemical effects observed in rats may not be the same as those observed in humans and that a different strain of rat may respond differently or be a more appropriate model for the cardiac effects of ibogaine in humans.

## 4. Materials and Methods

### 4.1. Animals

Wistar Han IGS rats bred in the kennel of the Institute for Biological Research “Siniša Stanković”—National Institute of the Republic of Serbia, University of Belgrade were used in this experiment. Three-month-old healthy male and female rats with b.w.s of 290–320 g and 175–250 g, respectively, were housed individually at 22 °C, day/night 12 h/12 h, with access to food and tap water ad libitum. At the time of treatment, all female rats were in the estrus phase of the estrous cycle, as determined by examination of a daily vaginal lavage [50].

### 4.2. Experimental Design

The experimental procedures and the dosages and time intervals were the same as in our previous experiments, which have been described in detail in our previous publications [10,14,15]. Ibogaine (ibogaine hydrochloride, PubChem CID: 197059, purity 98.93%, Remøgen, Phytostan Enterprises Inc., Montreal, QC, Canada) was dissolved in deionized water (dH_2_O) by vigorous vortexing (at a stock concentration of 2 mg/mL) and kept in the dark until use. Animals were separated by sex 21 days after birth and kept in separate cages (4 animals per cage) until the experiment began. Animals of each sex were randomly divided into five groups (6 animals per group) and treated at 9 am with a dose of ibogaine per os by gavage (1 or 20 mg/kg b.w.). All animals received 1 mL of fluid (dH_2_O or ibogaine solution of the appropriate concentration) per 100 g b.w. by gavage. The experimental groups were as follows: C—control, administered dH_2_O; L6—lower dose of ibogaine (1 mg/kg b.w.), sacrificed after 6 h; L24—lower dose of ibogaine (1 mg/kg b.w.), sacrificed after 24 h; H6—higher dose of ibogaine (20 mg/kg b.w.), sacrificed after 6 h; H24—higher dose of ibogaine (20 mg/kg b.w.), sacrificed after 24 h. At the end of experiment, the rats were sacrificed by rapid decapitation using a rodent guillotine.

### 4.3. Tissue Preparation

After decapitation, the hearts were removed and rinsed in physiological saline. The latero-caudal part of the left ventricular wall was removed and fixed with buffered 4% paraformaldehyde, pH 7.4, for 24 h for histological analyses. The remainder of the heart was frozen in liquid nitrogen and stored at −70 °C for biochemical analyses.

### 4.4. Histological Analysis

Fixed tissue samples were dehydrated in increasing concentrations of ethanol and xylene and then embedded in Histowax (Histolaboduct AB, Göteborg, Sweden). Thin sections (5 μm) were cut with a rotary microtome and stained with hematoxylin and eosin (H&E) and periodic acid-Schiff staining (PAS) [51]. PAS is used to stain glycogen granules in purple-magenta. A Leica DM2000 light microscope (Leica Microsystems, Wetzlar, Germany) with a Leica MC190 HD camera (Leica Microsystems) was used to analyze the histological preparations and to take the photomicrographs.

Possible pathologic changes were examined by H&E analysis of cardiac tissue that included histomorphologic examination of myocardium and pericardium. The presence of necrosis was determined by the visibility of nuclei and striation of cardiomyocytes. Necrosis was considered focal if less than 5% of the sample was affected, diffuse-moderate if 6–50% of the sample was affected, or diffuse-extensive if more than 51% of the sample was affected. Slides were also examined for myocardial mononuclear-cell infiltrate (indicator of myocarditis), perivascular inflammatory infiltrate (indicator of vasculitis), changes in the amount of adipose tissue, and the presence of pericarditis.

The amount of glycogen in cardiac tissue samples was assessed by quantitative and semi-quantitative analysis of PAS-stained slides, as described in our previous publications [10,14]. The determination of the number of glycogen-positive cells per 100 cells was performed at 400× magnification (objective magnification 40×), also known as high-power field of view (HPF). Samples were classified into one of three categories: less than 33.3%, 33.3–66.6%, or more than 66.6% glycogen-positive cells. Because glycogen-positive cells may contain varying amounts of glycogen, a semiquantitative analysis was performed by classifying each sample into one of three categories: + (weak staining), ++ (moderate staining), or +++ (strong staining), according to the average intensity of purple-magenta color in all fields examined. All slides were evaluated by three different investigators, and the score assigned by the majority is presented.

### 4.5. Measurement of the Activities of Antioxidant Enzymes, Glutathione S-Transferases (GST) and Xanthine Oxidase (XOD)

Frozen hearts were thawed, homogenized (3 × 10 s), and sonicated (3 × 15 s, at 10 kHz) in buffer containing 0.05 M tris(hydroxymethyl)aminomethane (Tris) and 1 mM ethylenediaminetetraacetic acid (EDTA), pH 7.4. A portion was separated for the measurement of TBARS and –SH group concentrations, and sucrose was added to the remainder at a final concentration of 0.25 M. Sonicates with sucrose were than centrifuged for 90 min at 105,000× *g*, at 4 °C (OptimaTM L-100 XP Ultracentrifuge, Beckman Coulter, Brea, CA, USA), and the supernatants were used for spectrophotometric measurement of enzyme activities using a Shimadzu UV-160 spectrophotometer (Shimadzu Scientific Instruments, Kyoto, Japan). The activities of total SOD and SOD2 were determined by the adrenaline method, monitoring absorbance at 480 nm [52]. A SOD activity unit (U) was defined as the amount of enzyme required for a 50% reduction in the rate of autoxidation of adrenaline. SOD2 activity was measured after inhibition of SOD1 by 4 mM KCN. SOD1 activity was calculated by subtracting SOD2 activity from total SOD activity. CAT activity was measured by monitoring H_2_O_2_ consumption at 230 nm [53]. GSH-Px activity was measured by monitoring NADPH consumption at 340 nm according to the modified assay described by Paglia and Valentine [54]. Tert-butyl hydroperoxide was used as the substrate. The activity of GR was measured by the method based on NADPH oxidation and GSSG reduction, monitoring the decrease in absorbance at 340 nm [55]. The activity of GST was measured by monitoring the increase in absorbance at 340 nm, using 1-chloro-2,4-dinitrobenzene and GSH as substrates [56]. XOD activity was measured by monitoring uric acid production at 292 nm in the presence of xanthine as a substrate [57]. Enzyme activities were calculated in U per mg of protein and then expressed as a percentage of the control value.

### 4.6. Measurement of Concentration of TBARS, Non-Protein –SH Groups and Free Protein –SH Groups

Sonicates prepared as described in the previous paragraph were centrifuged at 9000× *g* for 15 min (MiniSpin, Eppendorf AG, Hamburg, Germany), and the supernatants were used to measure TBARS content and total and nonprotein –SH groups. The intensity of lipid peroxidation was estimated by measuring the TBARS concentration [58]. Samples were mixed with equal volumes of 0.6% 2-thiobarbituric acid and incubated at 95 °C for 10 min. The absorbance at 532 nm was measured using a Shimadzu UV-160 (Shimadzu Scientific Instruments, Japan). TBARS concentration was calculated using malondialdehyde as a standard. Concentrations of total –SH groups were measured according to Ellman’s protocol, as optimized for a microtiter plate [59]. Samples were incubated for 10 min with Ellman’s reagent (3 mM 5,5′-dithiobis-[2-nitrobenzoic acid]) in 0.1 M potassium phosphate buffer (pH 7.3). Absorbance at 412 nm was measured with a Multiskan Spectrum spectrophotometer (Thermo Fisher Scientific Oy, Vantaa, Finland). To determine the concentration of non-protein –SH groups, proteins were precipitated with sulfosalicylic acid and the supernatant was used for reaction with Ellman’s reagent. The concentration of free protein –SH groups was calculated by subtracting the concentrations of non-protein –SH groups from the concentrations of total –SH groups. Concentrations of both TBARS and –SH groups were also expressed as a percentage of the control value.

### 4.7. Measurement of Protein Concentration

Protein concentrations in cardiac tissue samples were determined according to the Lowry method by measuring absorbance at 670 nm using a Multiskan Spectrum spectrophotometer (Thermo Fisher Scientific, Vantaa, Finland). Bovine serum albumin was used as a standard [60].

### 4.8. Data Analysis and Statistical Procedures

Statistical analyses of the results were performed according to the protocols described by Hinkle, Wiersma and Jurs [61]. The enzyme activities and concentrations of TBARS and –SH groups were either expressed as mean ± SEM or normalized to the mean of controls and expressed as percentage of control ± SEM. Differences between the control and ibogaine-treated groups were tested by one-way ANOVA followed by Tukey’s HSD post hoc *t*-test. The tests were conducted separately for male and female rats, using absolute values. Differences in ibogaine effects between male and female rats and differences between doses and exposure time were examined by three-way ANOVA (groups: ♂L6, ♂L24, ♂H6, ♂H24, ♀L6, ♀L24, ♀H6, ♀H24; factors: sex (S), dose (D) and time (T)) followed by Tukey’s HSD post hoc *t*-test using logarithmically (ln) transformed normalized values. The significance level for all statistical tests was *p* < 0.05.

## 5. Conclusions

In this work, we have shown for the first time that oral administration of ibogaine causes dose-dependent myocardial necrosis in male and female rats, in contrast to the previously described histopathologic changes in the livers and kidneys of both sexes, which we consider to be relatively mild and reversible. This finding points to necrosis as a mechanism of cardiotoxicity of ibogaine, which is one of the major side effects of ibogaine use in humans and is often fatal. Ibogaine exposure leads to changes in antioxidant enzyme activities and increased indicators of oxidative stress that are not consistent with necrotic processes. These effects are sex-specific, suggesting subtle differences at the molecular level. Therefore, it is not possible to draw a definitive conclusion with regard to the role of redox processes or oxidative stress in the occurrence of cardiotoxic necrosis after ibogaine administration.

## Figures and Tables

**Figure 1 ijms-25-06527-f001:**
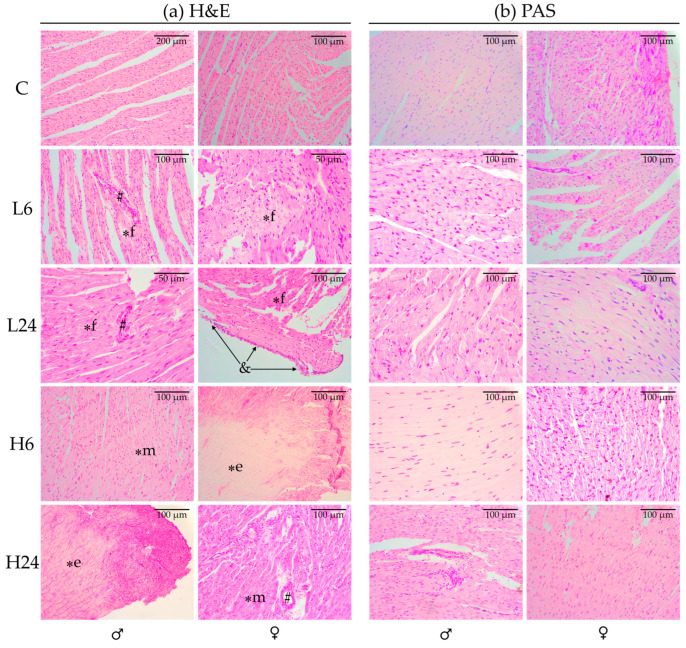
Micrographs of left ventricle wall sections from the latero-caudal part of the heart from male and female rats after per os treatment with ibogaine. Micrographs were taken using objective magnifications of 10×, 20×, or 40×. Scale bars are presented on each micrograph. The control group (C) was treated with dH_2_O; the other groups were treated with ibogaine, as follows: 1 mg/kg b.w., decapitated after 6 h (L6); 1 mg/kg b.w., decapitated after 24 h (L24); 20 mg/kg b.w., decapitated after 6 h (H6); 20 mg/kg b.w., decapitated after 24 h (H24). (**a**) H&E staining; Photomicrographs show normal structure of cardiac tissue in both control groups; myocardial necrosis–focal (∗f), diffuse moderate (∗m), and diffuse extensive (∗e); perivascular inflammatory infiltrate (#); pericarditis (&). Necrosis was classified as focal, diffuse moderate, or diffuse extensive if up to 5%, 6–50%, or more than 51% of the sample was affected, respectively. (**b**) PAS staining.

**Figure 2 ijms-25-06527-f002:**
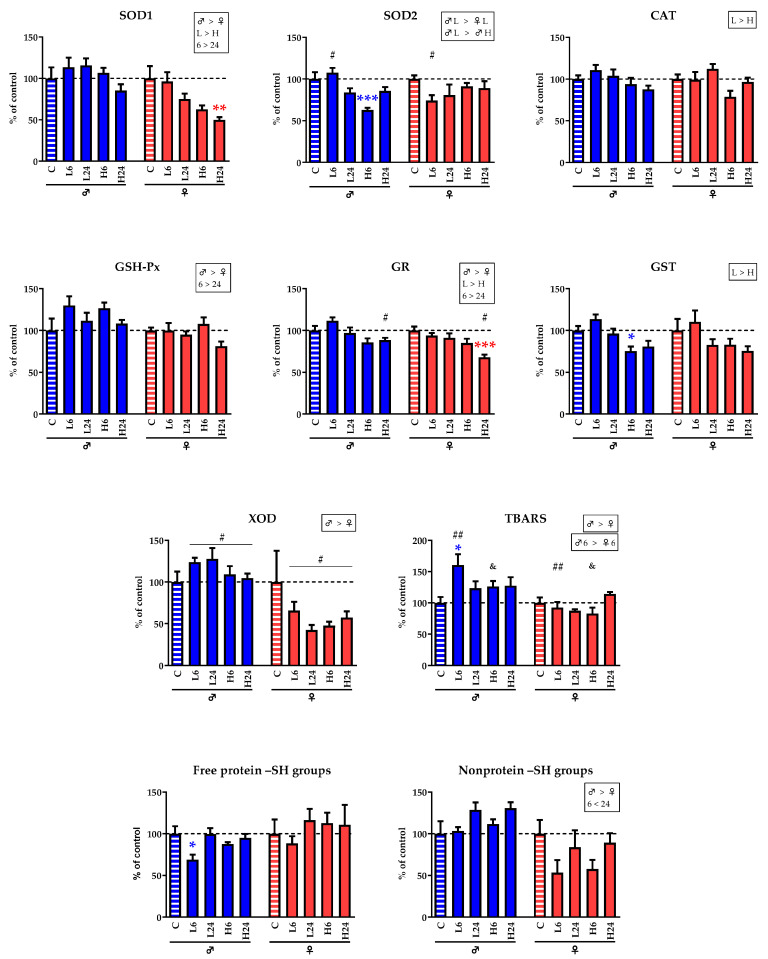
Ratio of changes in the activities of antioxidant enzymes (SOD1, SOD2, CAT, GSH-Px, and GR), glutathione S-transferases (GST), and xanthine oxidase (XOD) and the concentrations of TBARS, free protein –SH groups, and non–protein SH groups in the hearts of male and female rats treated with a single dose of ibogaine per os. The values shown are normalized relative to controls. The control group (C) was treated with dH_2_O; the other groups were treated with ibogaine as follows: 1 mg/kg b.w., decapitated after 6 h (L6); 1 mg/kg b.w., decapitated after 24 h (L24); 20 mg/kg b.w., decapitated after 6 h (H6); 20 mg/kg b.w., decapitated after 24 h (H24). The results of the analysis of variance are shown in Table 2. Blue and red asterisks (*) represent significant differences compared to the corresponding control group in male and female rats, respectively. Black symbols (# and &) represent significant differences between the individual experimental groups of the different sexes. Tukey’s HSD post hoc *t*-test: *, #, &—*p* < 0.05; **, ##—*p* < 0.01; ***—*p* < 0.001.

**Table 1 ijms-25-06527-t001:** Histological analysis of heart tissue (latero-caudal part of left ventricle wall) from male and female rats after per os treatment with a single dose of ibogaine: (**a**) Histopathological assessment of H&E-stained slides; (**b**) Assessment of glycogen on PAS-stained slides.

**(a) H&E**																					
	**Myocardial Necrosis**								**Myocardial Mononuclear** **Cell Infiltrate**			**Perivascular** **Inflammatory** **Infiltrate**			**Amount of Adipose Tissue**						**Pericarditis**
	**Absent**		**Focal**	**Difuse Moderate**		**Difuse** **Extensive**									**Decreased**	**Normal**		**Increased**			
** ♂C **	**6/6**		0/6	0/6		0/6			0/6			0/6			0/6	**6/6**		0/6			0/6
** ♂L6 **	0/6		**6/6**	0/6		0/6			0/6			**6/6**			0/6	**6/6**		0/6			0/6
** ♂L24 **	**1/6**		**5/6**	0/6		0/6			0/6			**6/6**			0/6	**6/6**		0/6			0/6
** ♂H6 **	0/5		0/5	**5/5**		0/5			0/5			0/5			0/5	**5/5**		0/5			0/5
** ♂H24 **	0/6		0/6	**3/6**		**3/6**			0/6			**1/6**			0/6	**6/6**		0/6			0/6
** ♀C **	**6/6**		0/6	0/6		0/6			0/6			0/6			0/6	**6/6**		0/6			0/6
** ♀L6 **	**3/4**		**1/4**	0/4		0/4			0/4			0/4			0/4	**4/4**		0/4			0/4
** ♀L24 **	0/4		**4/4**	0/4		0/4			0/4			**2/4**			0/4	**4/4**		0/4			**3/4**
** ♀H6 **	0/6		0/6	0/6		**6/6**			0/6			**5/6**			0/6	**6/6**		0/6			0/6
** ♀H24 **	0/6		0/6	**6/6**		0/6			0/6			**6/6**			0/6	**6/6**		0/6			0/6
**(b) PAS**																					
		**% of Glycogen Positive Cells**											**Intensity of PAS Staining**								
		**<33.3%**			**33.3–66.6%**		**>66.6%**						**+**				**++**		**+++**		
** ♂C **		**5/5**			0/5		0/5						**5/5**				0/5		0/5		
** ♂L6 **		0/4			**4/4**		0/4						**4/4**				0/4		0/4		
** ♂L24 **		**2/4**			**2/4**		0/4						**4/4**				0/4		0/4		
** ♂H6 **		**5/5**			0/5		0/5						**5/5**				0/5		0/5		
** ♂H24 **		**6/6**			0/6		0/6						**6/6**				0/6		0/6		
** ♀C **		0/6			**3/6**		**3/6**						**5/6**				**1/6**		0/6		
** ♀L6 **		0/2			**1/2**		**1/2**						**2/2**				0/2		0/2		
** ♀L24 **		**2/4**			**2/4**		0/4						**4/4**				0/4		0/4		
** ♀H6 **		**4/5**			**1/5**		0/5						**5/5**				0/5		0/5		
** ♀H24 **		0/6			**6/6**		0/6						**6/6**				0/6		0/6		

The control group (C) was treated with dH_2_O; the other groups were treated with ibogaine as follows: 1 mg/kg b.w., decapitated after 6 h (L6); 1 mg/kg b.w., decapitated after 24 h (L24); 20 mg/kg b.w., decapitated after 6 h (H6); 20 mg/kg b.w., decapitated after 24 h (H24). Necrosis was classified as focal, diffuse moderate, or diffuse extensive if up to 5%, 6–50%, or more than 51% of the sample was affected, respectively. Intensity of PAS staining was determined by classifying each sample in one of the three classes: + (weak staining), ++ (medium staining) and +++ (strong staining).

**Table 2 ijms-25-06527-t002:** Activities of antioxidant enzymes (SOD1, SOD2, CAT, GSH-Px, and GR), glutathione S-transferases (GST), and xanthine oxidase (XOD) and concentration of TBARS, free protein –SH groups, and nonprotein –SH groups in the hearts of male and female rats after per os treatment with a single dose of ibogaine.

			♂C	♂L6	♂L24	♂H6	♂H24		♀C	♀L6	♀L24	♀H6	♀H24
**SOD1**	**mean** **SEM** **n**		6.354 0.844 6	7.210 0.735 6	7.348 0.542 6	6.776 0.393 6	5.412 0.492 6		9.107 1.360 6	8.766 1.020 6	6.838 0.589 6	5.702 0.423 6	**4.540 **** **0.301** **6**
**SOD2**	**mean** **SEM** **n**		3.414 0.280 6	3.676 0.197 6	2.860 0.174 6	**2.142 ***** **0.098** **6**	2.924 0.162 6		3.169 0.141 6	2.347 0.211 6	2.566 0.406 6	2.890 0.126 6	2.824 0.260 6
**CAT**	**mean** **SEM** **n**		12.10 0.543 6	13.38 0.768 6	12.53 0.925 6	11.36 0.928 6	10.58 0.563 6		13.53 0.749 6	13.42 1.244 6	15.16 0.805 6	10.65 0.965 6	13.05 0.729 6
**GSH-Px**	**mean** **SEM** **n**		35.23 5.038 6	45.71 3.902 6	39.34 3.321 6	44.56 2.392 6	38.16 1.532 6		35.38 1.275 6	35.51 3.305 6	33.84 1.370 6	38.41 2.737 6	28.87 1.995 6
**GR**	**mean** **SEM** **n**		25.38 1.428 6	28.84 1.052 6	25.03 1.753 6	22.11 1.296 6	22.86 0.704 6		20.68 0.987 6	19.41 0.659 6	18.84 1.114 6	17.56 1.080 6	**14.02 ***** **0.674** **6**
**GST**	**mean** **SEM** **n**		98.06 5.343 6	111.3 5.416 6	94.46 5.714 6	**73.76 *** **5.481** **6**	79.10 6.815 6		85.58 11.81 6	94.43 11.57 6	70.80 5.836 6	71.01 6.126 6	64.59 4.845 6
**XOD**	**mean** **SEM** **n**		0.0433 0.0054 6	0.0536 0.0022 6	0.0552 0.0057 6	0.0472 0.0042 6	0.0452 0.0024 6		0.0485 0.0182 6	0.0319 0.0051 6	0.0205 0.0030 6	0.0231 0.0022 6	0.0278 0.0036 6
**TBARS**	**mean** **SEM** **n**		23.22 2.130 6	**37.24 *** **4.065** **6**	28.62 2.569 6	29.28 2.027 6	29.60 3.156 6		55.30 4.741 6	51.10 4.901 6	48.26 1.278 6	45.72 5.307 6	63.14 1.723 6
**Free protein –SH groups**	**mean** **SEM** **n**		200.7 18.29 6	**138.0 *** **12.37** **5**	200.0 14.06 5	176.0 4.256 5	190.5 9.886 6		81.01 13.81 6	71.65 6.962 6	94.09 11.05 6	91.32 10.18 6	89.66 19.39 6
**Nonprotein –SH groups**	**mean** **SEM** **n**		2.209 0.816 6	2.282 0.223 5	2.839 0.445 5	2.463 0.289 5	2.884 0.395 6		6.883 2.790 6	3.376 2.512 6	5.764 3.423 6	3.963 1.844 6	6.145 1.923 6

The control group (C) was treated with dH_2_O; the other groups were treated with ibogaine as follows: 1 mg/kg b.w., decapitated after 6 h (L6); 1 mg/kg b.w., decapitated after 24 h (L24); 20 mg/kg b.w., decapitated after 6 h (H6); 20 mg/kg b.w., decapitated after 24 h (H24). Tukey’s HSD post hoc *t*-test: *—*p* < 0.05; **—*p* < 0.01; ***—*p* < 0.001.

**Table 3 ijms-25-06527-t003:** Analysis of variance (one-way ANOVA) of the activities of antioxidant enzymes (SOD1, SOD2, CAT, GSH-Px and GR), glutathione S-transferases (GST), and xanthine oxidase (XOD), as well as concentrations of TBARS, free protein –SH groups, and nonprotein –SH groups in the hearts of male and female rats treated with a single dose of ibogaine per os: (**a**) male rats, absolute values; (**b**) female rats, absolute values.

**(a) One-Way ANOVA (♂C, ♂L6, ♂L24, ♂H6, ♂H24)**										
	SOD1	SOD2	CAT	GSH-Px	GR	GST	XOD	TBARS	Free protein –SH groups	Nonprotein –SH groups
	n.s.	F = 9.44 *p* < 0.001	n.s.	n.s.	F = 4.22 *p* < 0.01	F = 6.84 *p* < 0.001	n.s.	F = 3.00 *p* < 0.05	F = 3.79 *p* < 0.05	n.s.
**(b) One-Way ANOVA (♀C, ♀L6, ♀L24, ♀H6, ♀H24)**										
	SOD1	SOD2	CAT	GSH-Px	GR	GST	XOD	TBARS	Free protein –SH groups	Nonprotein –SH groups
	F = 5.46 *p* < 0.01	n.s.	F = 3.12 *p* < 0.05	n.s.	F = 7.56 *p* < 0.001	n.s.	n.s.	F = 2.95 *p* < 0.05	n.s.	n.s.

The control group (C) was treated with dH_2_O; the other groups were treated with ibogaine as follows: 1 mg/kg b.w., decapitated after 6 h (L6); 1 mg/kg b.w., decapitated after 24 h (L24); 20 mg/kg b.w., decapitated after 6 h (H6); 20 mg/kg b.w., decapitated after 24 h (H24). The results of Tukey’s HSD post hoc *t*-test are presented in Table 2. n.s.—non-significant.

**Table 4 ijms-25-06527-t004:** Analysis of variance (three-way ANOVA) of the activities of antioxidant enzymes (SOD1, SOD2, CAT, GSH-Px, and GR), glutathione S-transferases (GST), and xanthine oxidase (XOD), as well as concentrations of TBARS, free protein –SH groups, and nonprotein –SH groups in the hearts of male and female rats treated with a single dose of ibogaine per os—logarithmically (ln) transformed normalized values.

Three-Way ANOVA (♂L6, ♂L24, ♂H6, ♂H24, ♀L6, ♀L24, ♀H6, ♀H24)										
	SOD1	SOD2	CAT	GSH-Px	GR	GST	XOD	TBARS	Free protein –SH groups	Nonprotein –SH groups
**Factor:** **sex**	**♂ > ♀** F = 42.26 *p* < 0.001	n.s.	n.s.	**♂ > ♀** F = 18.32 *p* < 0.001	**♂ > ♀** F = 10.66 *p* < 0.001	n.s.	**♂ > ♀** F = 118.8 *p* < 0.001	**♂ > ♀** F = 31.41 *p* < 0.001	n.s.	**♂ > ♀** F = 24.03 *p* < 0.001
**Factor:** **dose**	**L > H** F = 20.32 *p* < 0.001	n.s.	**L > H** F = 11.37 *p* < 0.01	n.s.	**L > H** F = 23.25 *p* < 0.001	**L > H** F = 17.44 *p* < 0.001	n.s.	n.s.	n.s.	n.s.
**Factor:** **time**	**6 > 24** F = 6.49 *p* < 0.05	n.s.	n.s.	**6 > 24** F = 9.02 *p* < 0.01	**6 > 24** F = 5.66 *p* < 0.05	n.s.	n.s.	n.s.	n.s.	**6 < 24** F = 6.47 *p* < 0.05
**Interaction** **S × D**	n.s.	F = 14.47 *p* < 0.001	n.s.	n.s.	n.s.	n.s.	n.s.	n.s.	n.s.	n.s.
**Interaction** **S × T**	n.s.	n.s.	F = 5.19 *p* < 0.05	n.s.	n.s.	n.s.	n.s.	F = 5.43 *p* < 0.05	n.s.	n.s.
**Interaction** **D × T**	n.s.	F = 4.08 *p* < 0.05	n.s.	n.s.	n.s.	n.s.	n.s.	F = 6.73 *p* < 0.05	F = 4.33 *p* < 0.05	n.s.

Rats were treated with ibogaine as follows: 1 mg/kg b.w., decapitated after 6 h (L6); 1 mg/kg b.w., decapitated after 24 h (L24); 20 mg/kg b.w., decapitated after 6 h (H6); 20 mg/kg b.w., decapitated after 24 h (H24). The results of Tukey’s HSD post hoc *t*-test are presented in Figure 2. S—sex; D—dose; T—time; n.s.—non-significant.

## Data Availability

The data used to support the findings of this study are included within the article.

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
