# Peer review of "Ibogaine Induces Cardiotoxic Necrosis in Rats—The Role of Redox Processes"

_ijms, 2024, doi:10.3390/ijms25126527_

Round 1
Reviewer 1 Report
Comments and Suggestions for Authors
The authors investigated the effect of a single oral dose (1 or 20 mg/kg) of ibogaine on cardiac histopathology and oxidative/antioxidant balance. Ibogaine is an organic indole alkaloid that induces cardiac arrhythmias. Innovatively, ibogaine-induced dose-dependent cardiotoxic necrosis, which is not a consequence of inflammation, was found. Investigating the role of redox processes and oxidative stress in the occurrence of cardiotoxic necrosis after application of ibogaine is clearly and consistently tracked. The study of the free -CH groups is very appropriate. The statistical characteristics made are indicative and appropriate after conducting the research.
Small remarks :
1. I have no remarks about the introduction, materials and methods and discussion. In the discussion section, I suggest adding at least 6 additional references highlighting the research;
2. I suggest separating the results from the tables, too much information makes understanding the results difficult.The description under figures and tables is comprehensive.
3. the conclusion to be revised - does not fully express the significance of the study
4. Just12% of the used references are from the last 4 years. If possible to be replaced.
Comments on the Quality of English Language
Minor editing of English language required
Reviewer 2 Report
Comments and Suggestions for Authors
The manuscript Uzelac et al. is fascinating. The study is done with adequate biological replicates and controls. However, I have some questions:
1. Why do male and female rats respond differently to Ibogaine?
2. I do not think that it will be wise to exclude the changes in immune cell infiltration by looking at H&E staining only. If you were to make that statement you have to perform flow cytometric analysis for immune cells.
3. If only 24h of Ibogaine showed this much cardiotoxicity, I wonder what the survival rate for low-dose long-term Ibogaine treatment will be.
4. Could you please show the effect of Ibogaine only affects the heart and no other organs?
5. Could it be possible to add a cardiac function test (echo) to complement your findings?
Reviewer 3 Report
Comments and Suggestions for Authors
This study adds new knowledge regarding the cardiac effects of ibogaine using an established rat model. The manuscript is well-written, data are presented clearly and the authors have provided a comprehensive analysis of results based on their own previous findings and the reports of others. I recommend the manuscript for publication but I feel that there are several aspects of the study and its interpretation that require further clarification. These are listed below.
1. Please state the source of ibogaine in the Methods section.
2. Please state the source of Wistar rats and the exact Wistar rat strain used in the Methods section.
3. The authors do not mention animal ethics committee approval for the study. These details, including the name of the animal ethics committee that provided approval, should be added.
4. The types of wording used to describe approved methods for euthanasia of rats should be screened in the published literature and similar wording used to describe the protocol that the authors used in their study. The current description, “the rats were decapitated” is an insufficient explanation and may be interpreted by readers as an unethical practice.
5. A limitation of the study is that the cardiac responses noted may be directed by the rat strain used in the study and may not be able to be replicated in other rat strains (or provide a suitable model of the cardiac effects of ibogaine in humans). Because cardiac effects of ibogaine are poorly understood in humans, animal models are one of the best surrogates for study, however the caveat that pathological and biochemical effects observed in rats may not mirror that seen in humans should always be noted.
6. Although there are studies where ibogaine doses up to 20mg/kg have been used, including the authors’ previous studies, more recent clinical guidelines and clinical trials use and recommend 12mg/kg as the maximum dose. In these circumstances, the dose-dependent increase in cardiac pathology may not be as severe as some of the effects observed in the present study. At high ibogaine doses, the magnitude of cardiac abnormalities may not be direct effects of ibogaine but may in part be off-target effects.
7. The magnification and size of images depicting ibogaine-induced changes in cardiac muscle tissue are too small (and become pixelated when enlarged on screen). Could the authors please enlarge the magnification and size of images so that readers with expertise in these areas can examine them more thoroughly and to ensure that the effects that the authors describe are clearly visible?
8. In this study, rats were only kept alive for 24 hours following ibogaine administration. During this time, did the rats display any signs of ill-health? Based on the authors’ experience with this model, would the rats have continued to live normally (was this a dose of ibogaine that is known to cause the death or other health effects of all or a proportion of rats)? This is an important aspect to discuss as it would indicate whether the biochemical and/or histological effects that the authors describe are associated with deleterious health effects or death, or whether the effects are reversible or recoverable. Do the authors have or have access to cardiac tissues from rats who lived for longer periods of time following ibogaine administration that could be compared to tissues from the rats used in this study?
9. From reading the article, it appears that the authors have performed the study as a single experiment and that the repeatability of results is based on the fact that there were 6 animals per experimental group and not that results were similar when treatments were repeated using an independent group of rats. Is this correct? Some of the results seem quite ‘random’ in that effects are seen at a lower dose of ibogaine or at an earlier timepoint, but not at the higher ibogaine dose or later timepoint. While it may not be feasible to repeat the experiment at this stage, this is another limitation of the study design that should also be discussed. It is quite common that animal experiments may not produce the same results when repeated with a different batch of animals, however it is important that this limitation is acknowledged.
10. There are some more recent references and systematic reviews of ibogaine treatment that the authors could include. Some of the references included in the manuscript have been updated in more recent publications.
Comments on the Quality of English LanguageEnglish language is generally good, however there are some instances of incorrect word usage and grammatical errors that require correction.
Reviewer 4 Report
Comments and Suggestions for Authors
The manuscript titled “Ibogaine induces cardiotoxic necrosis in rats ‒ the role of redox processes” by Uzelac et al. investigates the impact of low and high doses of ibogaine, a natural product used in alternative medicine, following 6 and 24 h treatment on cardiac histopathology, antioxidant defenses, and markers of oxidative stress in rat hearts. Potential sex differences were also evaluated. The authors conclude that oral administration of ibogaine causes dose-dependent myocardial necrosis in male and female rats. Although the paper presents interesting and potentially important observations, there are some concerns.
1) Since the paper aims to explore the potential role of oxidative stress, ROS levels need to be assessed also. Although certain antioxidant defenses and lipid peroxidation was analyzed, no experiments were included to detect ROS.
2) Thioredoxin/thioredoxin reductase system (and peroxiredoxin 3) need to be evaluated also, especially considering their role in regulating mitochondrial H2O2 emission.
3) The authors include the potential role of ATP depletion in the discussion. The paper will be stronger if cardiac ATP levels are reported rather than only discussed.
4) The paper will be stronger if the potential contribution of apoptosis (or lack thereof) is included.
Minor:
Page 5 line 137-138: There seems to be a typo in “SOD1 and GR in the cardiac tissue of female rats treated with a higher dose of ibogaine were significantly increased after 24 h”.
Round 2
Reviewer 3 Report
Comments and Suggestions for Authors
The investigators have answered reviewers’ comments apart from a comment regarding more recent publications, which prompted quite an unusual response (for the reviewer to suggest references for the authors to evaluate!). Here is an example. There have been recent systematic reviews of the literature in this area; one of these is [Ona, G., Rocha, J.M., Bouso, J.C. et al. The adverse events of ibogaine in humans: an updated systematic review of the literature (2015–2020). Psychopharmacology 239, 1977–1987 (2022)]. The importance of recent and updated reviews is that they provide a more balanced background to the area, in particular to readers who may not be aware of the history of the subject. This is quite important as the investigators (and others who have published research articles in this area) frequently include statements such as “numerous cases of life-threatening complications and sudden deaths associated with ibogaine use have been reported”. The word “numerous” means different numbers to different people, and an important aspect of these “numerous” complications is that they haven’t occurred in the context of clinical trials or medical institutions where ibogaine purity and the medical history and monitoring of patients was documented. It would be distracting and unnecessary for the investigators to summarise all of this history in the present manuscript, however, inclusion of a very recent review (not an older review) such as the one suggested, will lead readers to more comprehensive summaries of the history of this area. I hope that this example adequately illustrates this reviewer’s comment to investigators and that they are able to proceed with the examination of references in their manuscript.
Comments on the Quality of English LanguageN/A
Reviewer 4 Report
Comments and Suggestions for Authors
The authors have mainly addressed my concerns. However, it should be pointed out that ROS levels can be detected in either freshly isolated or frozen cardiac tissue after sacrifice. The lack of these types of analyses should be at least included as a limitation.
